# "Fluctuation is the norm": Rehabilitation practitioner perspectives on ambiguity and uncertainty in their work with persons in disordered states of consciousness after traumatic brain injury

**Christina Papadimitriou**[1]*, **Jennifer A. Weaver**[2], **Ann Guernon**[3], **Elyse Walsh**[4],
**Trudy Mallinson**[5], **Theresa L. Bender Pape**[4,6]

1 Departments of Interdisciplinary Health Sciences, and Sociology, Oakland University, Rochester, MI, United States of America, 2 Department of Occupational Therapy, Colorado State University, Fort Collins, CO, United States of America, 3 Speech-Language Pathology Department, Lewis University, Romeoville, IL, United States of America, 4 Research Service and Center for Innovation in Complex Chronic Healthcare, Edward Hines Jr. VA, Hines, IL, United States of America, 5 Department of Clinical Research & Leadership, George Washington University, Washington, DC, United States of America, 6 Department of Physical Medicine and Rehabilitation, Northwestern University, Chicago, IL, United States of America

* cpapadimitriou@oakland.edu

**Data Availability Statement:** Per our funding sources, we are not allowed to share publicly these

## Abstract

The purpose of this study is to describe the clinical lifeworld of rehabilitation practitioners who work with patients in disordered states of consciousness (DoC) after severe traumatic brain injury (TBI). We interviewed 21 practitioners using narrative interviewing methods from two specialty health systems that admit patients in DoC to inpatient rehabilitation. The overarching theme arising from the interview data is "Experiencing ambiguity and uncertainty in clinical reasoning about consciousness" when treating persons in DoC. We describe practitioners' practices of looking for consistency, making sense of ambiguous and hard to explain patient responses, and using trial and error or "tinkering" to care for patients. Due to scientific uncertainty about diagnosis and prognosis in DoC and ambiguity about interpretation of patient responses, working in the field of DoC disrupts the canonical meaning-making processes that practitioners have been trained in. Studying the lifeworld of rehabilitation practitioners through their story-making and story-telling uncovers taken-for-granted assumptions and normative structures that may exist in rehabilitation medical and scientific culture, including practitioner training. We are interested in understanding these canonical breaches in order to make visible how practitioners make meaning while treating patients.

## Introduction

In this paper, we describe the clinical lifeworld in which rehabilitation practitioners work when treating persons remaining in states of disordered consciousness (DoC) after severe

data, therefore we submitted a file with all the de-identified data in this submission.

**Funding:** Funding for this project came from the Congressionally Directed Medical Research Programs (CDMRP) and Joint Warfighter Medical Research Programs (JWFMRP) of the United States Department of Defense. The funder had no role in the study design, data collection and analysis, decision to publish, or preparation of the manuscript. The specific award names and numbers are: CDMRP W81XWH-14-1-0568 and JWFMRP W81Xwh-16-2-0023.

**Competing interests:** No competing interests exist.

traumatic brain injury (TBI) [1–6]. About 59% of persons who receive specialty rehabilitation will recover from DoC within the first year of recovery [7]. For those remining in DoC recovery will continue for several years, but the odds of substantive recovery incrementally decreasing each year thereafter [7].

Recovery from DoC is described by a gradient of consciousness where less consciousness is associated with more disruption of functional and structural neural connectivity [8–21]. While the gradient is delineated clinically as the vegetative state (VS), minimally conscious state (MCS), and emergence from MCS (eMCS) recovery [22–25], is not necessarily a linear progression along this gradient [13, 26–28]. Persons remaining in states of DoC, often experience fluctuating levels of wakefulness and external awareness [29, 30] and, even with highly specialized care [31], this inherent variability in neurobehavioral performance obscures clinical observations of functioning during rehabilitation. This fluctuation challenges practitioners' day-to-day work because it is hard to unequivocally determine patients' level of consciousness. In this way, practitioners work in a context of scientific uncertainty regarding accurate detection of changes in levels and states of consciousness, which is the basis for monitoring recovery. At the same time, there is a lack of empirical data to guide clinical treatment [32], which creates ambiguity about treatment decision-making. How practitioners perceive and make sense of diagnostic and prognostic uncertainty and therapeutic ambiguity remains an uncharted psychosocial domain [33] and an unappreciated aspect of DoC rehabilitation treatment [34].

We report the ways practitioners provide rehabilitation services in spite of the day-to-day uncertainty and ambiguity. We also report the way practitioners talk about fluctuations in patient behavior through story-telling and story-making to make sense of patient recovery and treatment decisions. In addressing the uncharted domain of understanding how practitioners manage the uncertainty and ambiguity intrinsic in their day-to-day practice, this paper takes a non-traditional stance for rehabilitation science. Rather than positioning this study as an examination of how practitioners' clinical decision-making influences patient outcomes in order to inform quality improvement initiatives or advance person-centered care, we describe the more elementary practice of decision-making processes and underlying reasoning that enable monitoring of recovery and the inter-related treatment decisions. Doing so enables a more nuanced understanding of practitioners' everyday clinical reasoning and the strategies they use to cope with diagnostic uncertainty and therapeutic ambiguity. This is turn can lead to new insights and innovation regarding clinical practice and knowledge in sTBI and DoC.

## Epistemological underpinnings

Using hermeneutic and narrative approaches, we posit that "meaning and the processes by which meanings are created and negotiated within a community" form culture [35–40]. Making sense (i.e., the act of interpretation) is a fundamental part of the human condition and provides the basis to understand patient recovery and make treatment decisions.

Inpatient rehabilitation culture is dominated by medical and evidence-based scientific models in which practitioners are treated as experts who know what to do and can diagnose and prognosticate with confidence. This culture is driven by the positing of theory-driven, empirically-proven, measurable outcomes-based clinical practices, where patient recovery is the ideal outcome. As Mattingly et al note, "Culture gives us the possibility of reading other minds because a cultural world is one where meanings are public and communal, rather than individual and private" [41]. The culture of inpatient rehabilitation provides a foundation or canon by which practitioners can make sense of their work, and guides actions when canonical breaches or violations occur [35]. Practitioners in DoC work within a world of canonical

breaches due to the combined scientific uncertainty [32] about diagnosis and ambiguity about treatment decisions in the face of fluctuating patient behaviors, which disrupt the canonical meaning-making processes in which practitioners are trained and in which health organizations operate. We are interested in understanding these canonical breaches and disruptions that exist within rehabilitation culture as they are exhibited in the stories practitioners in DoC told because they expose taken-for-granted assumptions and normative cultural structures that otherwise remain invisible. To do so, we used insights from the traditions of narrative medicine, grounded theory, and the first author's phenomenological training [1, 4, 5, 42].

Story-telling and therapeutic emplotment play important roles in rehabilitation practitioner sense-making [35, 41, 43]. In narrative interviews, we asked practitioners to tell us about times when interactions with patients were frustrating, surprising, or memorable (exciting, impactful, strange). These interview questions enabled practitioners to share their "stories from the field" [44], allowing us to see them as actors, even protagonists, in their story-telling, story-making, and meaning-making as they treated patients. These stories are how practitioners make sense of their day-to-day work. They are stories of unexpected patient responses, practitioners' explorations, improvisations, and successes. While we didn't design our study to focus on uncertainty and ambiguity in clinical practice, our epistemological approach allowed us to bring to the surface the ways practitioners in DoC make sense of the challenging interpretive process of treating patients.

## Methods

This study is nested within a larger clinical trial (NCT02366754) examining neurobehavioral, neural and molecular responses to repetitive transcranial magnetic stimulation provided to patients with DoC after severe TBI. This qualitative study aimed to advance understanding of how rehabilitation practitioners understand and communicate neurobehavioral change of these patients.

Collectively, the authors have many years of experience working with these patients and their families. The team includes occupational therapists, speech language pathologists, physical therapists, a phenomenological sociologist, and caregivers.

### Data collection

We conducted in-person interviews with 21 rehabilitation practitioners in two health systems from multiple rehabilitation disciplines (e.g., medical doctors, nurses, social workers, occupational therapists, physical therapists, and speech language pathologists) who each had at least six months experience working with patients with DoC after TBI. We used a purposive sampling strategy as our objective was to hear from practitioners who work in specialized DoC programs since they would have multiple patient encounters to reflect upon during interviewing. This is a common strategy in qualitative research designs where the goal is to gain in-depth understanding [36, 45] of process-oriented phenomena such as understanding behavioral change and meaning-making [37].

"Interviews are speech events that produce narratives that are jointly constructed by interviewers and respondents" [46]. Two rehabilitation practitioners (EW and AG) with expertise treating DoC patients conducted the interviews. They were trained by the first author, an expert in qualitative interviewing. All interviews were audio-recorded and transcribed verbatim. Institutional Review Board approvals were obtained from Northwestern (NU IRB: STU00203840) and Edward Hines, Jr. Veteran Affairs Hospital (Hines IRB: 16–037). Participants were provided with information letters about the study and verbally consented.

## Data analysis and reflexivity

We are aware that these data are "our own constructions of other people's constructions of what they and their compatriots are up to" [47]. Using the principles of grounded theory and narrative analysis, we analyzed interviews into major topics and themes based on participants' direct quotes [1, 2, 6, 48]. Our analyses were inductive in that we did not apply *a priori* constructs or theories to coding. We began with open, line-by-line coding [1, 48]. Three members of our team (JW, AG, CP) coded separately and discussed codes during weekly meetings. Each member read transcripts a minimum of three times.

Data collection and analysis occurred simultaneously and iteratively. We created categories from codes and memos [1, 48]. For this study, categories served as organizing 'buckets' including multiple codes that described similar topics or experiences and which later became themes. Codes and categories were not mutually exclusive, rather one code might fit within two categories. Themes typically involved combining more than one category (Fig 1). We developed a codebook to organize and appraise our decisions, and engaged in constant comparison analysis of analytic themes [1, 48]. We used the qualitative software NVIVO 11 to organize our data and work on codes and categories.

We practiced reflexivity, [49, 50] that is, critical awareness of our own positionality, biases, and emotions regarding the data, by writing personal and analytic memos [1]. We engaged in member-checking with the practitioners in our team by practicing "dialogical intersubjectivity" [1, 50], in which we exchanged positive and challenging emotions and thoughts related to the data, practiced active listening of each other's perspectives, and challenged each other to acknowledge our unique perceptions and predispositions (including personal, disciplinary, and professional). We documented these discussions in minutes to audit our decisions. When we disagreed, we re-read the transcripts until we reached simple group consensus [1]. In spite of all the careful work to not impose our experiences onto the data, it is possible that we have highlighted findings that are meaningful to us because of our experiences.

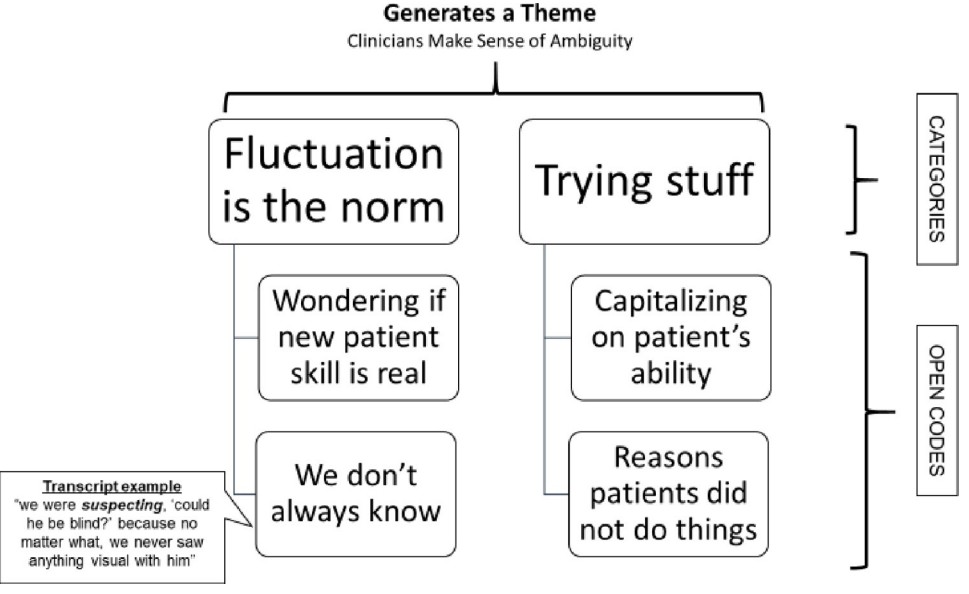

**Fig 1. Example of methodological approach to generating themes.**

# Findings

Participants were recruited and enrolled from two North American, mid-western clinical settings—a civilian (n = 7) and a veteran inpatient rehabilitation facility (n = 14) (Table 1). The majority of participants were female (n = 20), rehabilitation therapists (n = 13), with more than twenty years' experience in their profession (n = 11), and more than 5 years' experience working specifically with patients with DoC (n = 11). In an effort to preserve participants' anonymity, when quoting, we identify them with their professional designation (e.g., PT for Physical Therapist) followed by a number (e.g., PT 2). As is common in most qualitative reporting traditions, we do not enumerate how many participants agreed or mentioned a particular topic or category. Participant quotes are chosen because they best represent the themes we report in this paper. Participant characteristics are representative of the rehabilitation workforce. The American Speech-Language-Hearing Association (ASHA) reported in 2019, that 96% of the 175,000 SLPs certified by ASHA are female [51]. The results of the 2017 National Nursing workforce survey, indicate 90.9% of the RN workforce in the United States identifies as female [52]. The American Occupational Therapy Association Workforce and Salary Survey of 2019 reports that 91% of the OT workforce identifies as female [53]. Similarly, statistics published by the American Physical Therapy Association indicate that 65% of physical therapists and 71% of physical therapy assistants identify as female [54].

**Table 1. Participant demographics.**

| Demographic Information | | Number of Participants |
|---|---|---|
| Setting | Veteran Facility | 14 |
| | Civilian Facility | 7 |
| Discipline | Occupational Therapist | 4 |
| | Physical Therapist | 4 |
| | Speech and Language Pathologist | 3 |
| | Nursing | 2 |
| | Physical Medicine & Rehabilitation Medical Doctor | 2 |
| | Psychologist | 2 |
| | Recreational Therapist | 2 |
| | Certified Nursing Assistant | 1 |
| | Social Worker | 1 |
| Gender | Female | 20 |
| | Male | 1 |
| Age | 25–35 | 9 |
| | 36–45 | 4 |
| | 46–55 | 6 |
| | >55 | 2 |
| Years Practicing in Profession | <5 | 5 |
| | 5–10 | 3 |
| | 11–20 | 7 |
| | More than 20 | 11 |
| Years Practicing with Patients in DoC | 0–5 years | 9 |
| | 6–10 | 3 |
| | More than 10 | 9 |

## Experiencing ambiguity and uncertainty in clinical reasoning about consciousness

Ambiguity (inexactness) and uncertainty (unpredictability) are related. Ambiguity may arise in situations where multiple persons have differing interpretations of the same experience. In DoC, ambiguity arises due to imprecision of clinical assessments to document states of consciousness, multiple expert interpretations of patient states of consciousness, and uncertainty arises with limited evidence regarding efficacy of chosen treatments [32]. A common aspect of ambiguity is experiencing lack of confidence because there is imprecise or unknown information which render situations difficult to be sure about. These epistemic limitations make a patient's states of consciousness largely unknown. In the context of DoC, uncertainty exists due to individual patient variation in responses to treatment and fluctuating patient response. As such, detecting and determining signs of consciousness in patients in DoC is challenging. Practitioners made sense of patients' ambiguous signs of consciousness via patient stories whose leitmotif was 'looking for a person' in the patient in DoC' during treatment [55]. In other words, practitioners in DoC observe signs that can point to patients' intention, motivation, or volition that could not be classified as mere bodily reflexes or responses [56]. 'Looking for a person' was one of the ways practitioners talked about searching for consciousness when treating their patients.

We identify two major categories to describe ambiguity (inexactness) and uncertainty (unpredictability) when treating persons in DoC: "Fluctuation is the norm" and "Trying stuff" (Table 2). The first describes practitioners' experiences of clinical reasoning about diagnosing patients' levels or states consciousness by searching for consistency and making sense of ambiguous patient responses to describe their recovery. The second describes what practitioners do in spite of uncertainty in the face of paucity of empirical evidence and brings to the surface that practitioners go outside their canonical training in order to make treatment decisions. Both categories represent ways in which practitioners make meaning and clinically reasons about patients' consciousness in the midst of ambiguity and uncertainty regarding treatment decisions.

**Clinical reasoning about consciousness when "fluctuation is the norm".** Fluctuation, variability, and lack of consistency were expressions all rehabilitation practitioners used to describe and interpret the recovery process of patients with DoC. One participant, PT7, elegantly captures this experience: "We expect fluctuation in this patient population. Fluctuation is the norm. We don't expect consistent performance." "Fluctuation is the norm" resonated with practitioners in our research team, and is echoed in the DoC literature [29, 30, 57]. PT8 similarly described: "I had a patient in our emerging consciousness program, he was making some gains but still wasn't consistently following commands. He was here for 12 weeks and for the majority of that time he was kinda at a similar level and it was just very variable. One day he seemed to be occasionally responding or doing things more consistently, while other days he was doing nothing."

When patients' responses are inconsistent and fluctuating, it is challenging for practitioners to be certain about how to interpret them. For example, are patients improving or deteriorating? In which state of consciousness do their responses best fit? PT7 wonders about her patient: "You know, is she [patient] consistent? Is she *truly* consistent? Like *100%* consistent? Or is she still inconsistent enough where you'd say she was technically still minimally conscious? Or had she *truly* emerged into that conscious state?" (Italics denotes emphasis in the audio transcript). Wondering about patients' recovery, rather than being confident in their assessment of patient progress or decline, characterized practitioners' ways of talking about their work.

**Table 2. Codebook example from the main theme of 'ambiguity and uncertainty among rehabilitation practitioners in DoC.**

| Main categories supporting theme | Supporting Subcategories (Description) | Participant Quotes | Profession, Participant # |
|---|---|---|---|
| **"Fluctuation is the norm"**<br><br>Describes practitioners' experiences of clinical reasoning about diagnosing patients' levels or states consciousness by searching for consistency and making sense of ambiguous patient responses to describe their recovery | | "We expect fluctuation in this patient population. Fluctuation is the norm. We don't expect consistent performance." | Physical Therapist, 7 |
| | | "I had a patient, he was in our emerging consciousness program, and he was making some gains but still wasn't consistently following commands . . . he was here for 12 weeks and for the majority of that time he was kinda at a similar level and . . . it was just very variable. One day he seemed to be occasionally responding or doing things more consistently, while other days he was doing nothing." | Physical Therapist, 8 |
| | *Searching to observe consistent* responses to stimuli as indications that the patient is improving | "You know, is she consistent? Is she *truly* consistent? Like 100% consistent? Or is she still inconsistent enough where you'd say she was technically still minimally conscious? Or had she truly emerged into that conscious state?" | Physical Therapist, 7 |
| | | "I can remember when [patient] would follow a command for the first time. [I thought] 'Whoa, did they actually just do that? Did I actually just see that? Or was that sort of random?'" | Physical Therapist, 7 |
| | | "What I typically like to see when I'm following patients is, you know, that they are beginning to show some localized and purposeful activity. We might start to see first some sort of intentional motor cognitive behavior and then that that's consistent. You're seeing that consistently and then that's kind of building into even more than that. Either following a command, like a yes/no or whether that's nodding or thumbs up or thumbs down. So something consistent." | Physiatrist, 16 |
| | *Collaborating* with others in teams to identify consistency of responses thus documenting level of consciousness | "as a team we talk about [possible change] and I'll say 'I am seeing a localized response, they are localizing to this' . . . and speech may say 'I see that but I'm not seeing it consistently.' So that is why sometimes we will wait until it's consistent cross disciplines before we jump between levels [of consciousness.]" | Occupational Therapist, 6 |
| | *Observing nuances* in patient responses, and *grappling with* unexplained recoveries or stalls in patient progress. | "is more of a gradual thing. I don't feel like one day you walk in and they are emerged . . . we're doing serial daily exams on the person and nursing is getting a 24/7 view. You have all this information that you are gathering all the time; so in my experience I would say it's more of taking all of that input in and it's not a black and white thing." | Physiatrist, 16 |
| | | "I feel like it's usually a pretty long and slow process and [patients] go from kind of a vacant stare with no recognition and no following or tracking movements. Usually the first thing we see is some sort of eye contact, some sort of effort to follow an object, or just pulling away if you touch them, or if you put your hand in their hand and they respond in some way with a hand movement. Usually, those are kind of the first signs that I start to notice." | Recreational Therapist, 11 |
| | | "Mr. Jones was our worst-case scenario patient. We, maybe, expected that he might regain some small level of function; and [yet] he's functioning on a level that no one can explain." | Recreational Therapist, 11 |
| | | "He never tracked in any way, he never focused on anything. At one point we were suspecting, 'could he be blind?' Because no matter what, we never saw anything visual with him." | Speech Language Pathologist, 1 |
| | | "[Patient] is really doing well from a physical perspective; much beyond my initial expectations were. So, it was actually a really good learning case for me because I thought I knew a lot at that point in my career and it was a good reminder to me of the things we don't always know." | Physical Therapist, 7 |

*(Continued)*

**Table 2.** (Continued)

| Main categories supporting theme | Supporting Subcategories (Description) | Participant Quotes | Profession, Participant # |
|---|---|---|---|
| **"Trying Stuff"** Describes what practitioners do in spite of uncertainty in the face of paucity of empirical evidence and brings to the surface that practitioners go outside their canonical training in order to make treatment decisions | | "I was trained by my colleagues to just try stuff. Because there is a lack of research with disorders of consciousness as far as interventions that actually work. A lot of the times I feel like we are trying stuff, and we are just [waiting to] seeing what happens." | Occupational Therapist, OT4 |
| | | "So my intern went 'stop, collaborate,' and he stopped and the patient mouthed the word 'listen'. We didn't hear anything at that time, but as we continued on with the song, [the patient] would finish the sentence and gradually we started to actually hear him verbalize the right word. So, we had tried everything, including songs that his wife said he liked. He didn't respond to those, but this was a song that he would have known as a young teenager, like 12 or 13 years old. And so somehow it stirred something different." | Recreational Therapist, 11 |
| | | "The patient's head was down and he wasn't making any eye contact or an effort to raise his head. And when the dog came in, we had to cue him to look, and then he raised his head and his eyes widened and he started to smile. And then when the dog came closer to him, he leaned in towards the dog more and when we put his hand on the dog's head, we saw him moving his fingers as if he was trying to scratch. He wasn't able, at that point, to reach purposefully to do it, but when we put his hand in place, he moved his fingers. His sustained attention was longer when the dog was there; I could get him to really focus for ten to fifteen minutes." | Physical Therapist, 6 |
| | | "I'm a very non-traditional, sort of out of the box therapist, and sometimes what these young males respond to is not necessarily a clinically standard and appropriate type of approach. There's a TV show called "Jackass" where these guys do ridiculous things and oftentimes they're just gross and inappropriate and in every way unacceptable behavior. But, I get a better response from "Jackass" than I do almost anything and so I put it on for this young man. . . . The first thing that I noticed, he was watching the screen and not just sitting there, you know, just unaware. He was focusing on the screen and he smiled at an appropriate time. So he recognized that the moment was funny and he smiled at the right time; and so that was my first, I guess, sign that he was starting to emerge." | Recreational Therapist, 3 |
| | | "There was singing, there was praying, there was shaking of rattles and drums and things like that. There was two people working with the patient and then two people that worked with his wife. They did breathing work with the wife to release emotional stuff and they did some massage. There was prayers in the Christian tradition and prayers in the Mayan tradition. Overall, it was a very emotional and amazing experience. The patient had been here for months and had no real response that we could see. So, immediately after that experience, he kind of went into this even deeper sleep, it was like he was knocked out for three days and on the third day when he woke up, he was present. His eyes had changed. He was tracking and showing responsiveness and he just went on this remarkable recovery process that nobody here can explain it. People talk about it and nobody has an explanation. People say it was, he was a miracle." | Recreational Therapist, 11 |
| | | "we need to try and stimulate [patient's] level of alertness in any way we can." | Physical Therapist, 7 |

PT7 exemplifies her search for consistency by repeating: "is she consistent", "*truly* consistent", "*100% consistent*." Her words parallel the specialized language of commonly used clinical assessment tools (such as the CRS-R [58] or Coma Near Coma Scale [59, 60]) where patient responses are scored according to consistency, defined by consensus of practitioners, and serves as a clinical marker for recovery [61, 62]. Seeking consistency where "fluctuation is the norm" points to efforts to cope with the ambiguity inherent in practitioners' work.

Practitioners stated the experience of a 'double take' when patients responded to a command for the first time. "I can remember when [patient] would follow a command for the first time. [I thought] 'Whoa, did they actually just do that? Did I actually just see that? Or was that sort of random?'" (PT7) A double take is a behavioral response to the cognitive dissonance practitioners may experience when treating these patients whose responses fluctuate so much and are inconsistent [63]. It may also be an example of doubt—"Did I actually just see that?" perhaps indicates not trusting one's own senses.

We turn next to examples of how practitioners make meaning in this treatment environment.

*Examples of meaning-making when "fluctuation is the norm".* Fleming and Mattingly assert that practitioners "simultaneously observe, assess, and interpret patient's actions" and this "thinking in action" is tacit and involves experiential knowledge (i.e. disciplinary and practical training, prior experience with patients, and astutely observing during treatment sessions) [37]. Practitioners make meaning as they act out their reasoning in treatment sessions with patients. The process of making meaning is active and creative, even though it is tacit. We show examples of practitioners' meaning-making and clinical reasoning when "fluctuation is the norm" around four themes: *searching to observe consistent* responses to stimuli as indications that the patient is improving, *collaborating* with others in teams to identify consistency of responses thus documenting level of consciousness, *observing nuances* in patient responses, and *grappling with* unexplained recoveries or stalls in patient progress.

P16 explains a clinical reasoning logic she uses in her searches for *consistency*: "What I typically like to see is, you know, that they are beginning to show some localized and purposeful activity. We might start to see first some sort of intentional motor cognitive behavior and then that that's consistent. You're seeing that consistently and then that's kind of building into even more than that. Either following a command, like a yes/no or whether that's nodding or thumbs up or down. So, something consistent."

Clinical reasoning in inpatient rehabilitation is not just an individual practitioner's process, it is *collaborative* and team-based. Deciding whether patients were consistent was discussed often in team meetings. OT6: "as a team we talk about [possible change] and I'll say 'I am seeing a localized response, they are localizing to this'. . . and Speech may say 'I see that but I'm not seeing it consistently.' So that is why sometimes we will wait until it's consistent cross disciplines before we jump between levels [of consciousness.]" Multiple practitioners need to agree that indeed the patient is responding consistently in order to formally document a state of consciousness.

In order to determine whether patients are responding consistently, practitioners described the importance of *observing* fine *nuances* in patient responses as an important clinical reasoning practice. Recovery "is more of a gradual thing. I don't feel like one day you walk in and they are emerged. . . . we're doing serial daily exams on the person and nursing is getting a 24/7 view. You have all this information that you are gathering all the time; so, in my experience I would say it's more of taking all of that input in and it's not a black and white thing." (P16) The action of "taking it all in" is a form of clinical reasoning practitioners engage in to look for consistent indications of recovery of function and consciousness. Noticing is an important observational tool to achieve this: "Usually the first thing we see is some sort of eye contact,

some sort of effort to follow an object, or just pulling away if you touch them, or if you put your hand in their hand and they respond in some way with a hand movement. . . . those are kind of the first signs that I start to notice." (RT11)

Though practitioners notice fine nuances and "take all the information in," they sometimes *can't explain* patients' recovery using formal clinical assessment information or their own expertise and scientific training. RT11: "Mr. Jones was our worst-case scenario patient. We expected that he might regain some small level of function; and [yet] he's functioning on a level that no one can explain." SLP1 shares an example of a patient whose assessment information showed "no response," but that didn't stop the clinical team from wondering why that was the case: "He never tracked in any way, he never focused on anything. At one point we were suspecting, 'could he be blind?' Because no matter what, we never saw anything visual with him." When clinical assessment information does not satisfy explanations for why a patient isn't responding to stimuli, plausible wondering may be a way of grappling with ambiguity.

While analyzing these data, we wondered: What do practitioners do with unexplainable recoveries? How do practitioners work with clinical information they can't explain? PT7 sees a chance to learn: "[Patient] is really doing well from a physical perspective; much beyond my initial expectations were. So, it was actually a really good learning case for me because I thought I knew a lot at that point in my career and it was a good reminder to me of the things we don't always know."

Clinical reasoning takes place during the act of treating patients, it is not a purely cognitive, thinking process. It is "thinking in action" that practitioners engage in when they provide different stimuli to observe nuances of behavior, collaborate with team members to better understand patient responses, and make sense of what they are observing in the moment to assess of patients' recovery status. During this "thinking in action," practitioners make meaning using prior knowledge and experience, assessing-in-the-moment information, and by comparing their observations of present behavior with patients' past performance. We turn next to explore further practitioners' "thinking in action" through practitioners' patient stories shared during interviews.

**Trying to find consciousness by "trying stuff".**   Through practitioners' patient stories we can learn how practitioners "think in action" and what they do during clinical sessions to evaluate patients' consciousness status, elicit responses, facilitate consistency in responses to various stimuli associated with a particular state of consciousness, or generate emerging responses for the next state of consciousness [62, 64, 65]. Through these stories we learn how they make sense of their interactions with patients' inherently fluctuating and inconsistent responses. "Thinking in action" involves trial and error. OT4: "I was trained by my colleagues to just try stuff. Because there is a lack of research with disorders of consciousness as far as interventions that actually work. A lot of time I feel like we are trying stuff, and we are just [waiting to] see what happens." Working in a clinical environment where practitioners "try stuff" and wait to "see what happens" is unlike other rehabilitation fields where recovery trajectories are more predictable and there is less clinical equipoise.

"Trying stuff" and "seeing what happens" frame the stories practitioners told us. Their stories communicate more than strategies or tools they use to evaluate and treat. We see the creation and enactment of plots that help organize their observations and give meaning to unfamiliar or hard to explain situations. The uncertainty of responses to treatments and recovery trajectories for persons with DoC is a breach or challenge to the canonical scientific reasoning practitioners are trained in and comfortable with; it is no surprise that they share stories in which they narrate how they make sense of challenging interactions with patients. In narrating, they tell sense-making stories of complex or impactful situations, and position themselves

as actors, even protagonists. In a medical culture where practitioners are supposed to know what to do and how to do it, treating patients in DoC may disrupt these suppositions.

In the examples that follow, we use practitioners' stories of patient interactions that focus on the theme of "trying stuff". In these stories, practitioners use music, video, prayer, and a dog to elicit responses and treat patients. They cast themselves sometimes in the role of explorer, improviser, or rebel, and they tell stories of miracles, informed experimentation, and lucky happenstances. In these stories, practitioners become tinkerers [66–69].

*"Trying stuff:" Examples of "thinking in action".* RT11 used music to elicit responses from a patient who was alert and used hand gestures but was not verbalizing. RT11 works alongside a young and energetic male intern, who sang a popular song by Vanilla Ice called 'Ice, Ice, Baby'. The refrain is 'stop, collaborate, listen':

"[the intern] sang 'stop, collaborate,' and stopped, and the patient mouthed the word 'listen'. We didn't hear anything at that time, but as we continued on with the song, [the patient] would finish the sentence and gradually we started to actually hear him verbalize the right word. We had tried *everything*, including songs that his wife said he liked. He didn't respond to those, but this was a song that he would have known as a young teenager, like 12 or 13 years old. And so somehow it stirred something."

What can't be read in this passage is the excitement in the practitioner's voice about the increasing consistency of and improvement in the quality of elicited responses, starting first with mouthing and then verbalizing the song words. In this story, RT11 explains some of the reasoning strategies clinicians use including gathering information from individuals with close knowledge of patient preferences, such as the patient's wife, and also the in-the-moment lucky serendipity of a song sung by a team member. In declaring they had tried "everything," RT11 is acknowledging the interplay of clinical judgement and guesswork/ trial and error aspects in treatment planning. Sound clinical choices, such as the patient's past preferred music, are supplemented with in-the-moment lucky happenstance. RT11 exemplifies one way of "thinking in action": building implicit, individualized theories to explain what worked or didn't work with patients.

Another instance of practitioners "trying stuff" comes from a collaboration with family to bring a dog to a patient's room [70]. PT6 told us, "The patient's head was down and he wasn't making any eye contact or an effort to raise his head. When the dog came in, we had to cue him to look, and then he raised his head and his eyes widened and started to smile. When the dog came closer to him, he leaned in towards the dog more and when we put his hand on the dog's head, we saw him moving his fingers as if he was trying to scratch. His sustained attention was longer when the dog was there; I could get him to really focus for ten to fifteen minutes." In this example, the practitioner is reflecting on how the patient's attention when the dog is present is longer and more sustained than prior sessions without the dog. In this brief story, we see the practitioner making sense of the patient's improvement (more consistent, sustained attention) as being related to the presence of a dog (with whom he felt connected). The practitioner explored bringing a dog in treatment as part of "trying things" and now builds her own knowledge base of possible interventions that might work with these patients.

Practitioners operate with few validated approaches in their treatment toolbox. As a result, they perceive their informed experimentation as radical and norm-breaking. Yet, in reality, normative rehabilitation practice in this area offers little guidance since recovery is unpredictable and the tool box of options for treatment with established efficacy are limited. In the next example, RT3 casts herself in the role of a rebel, or acting 'outside the box' to tell a story of non-conformity and success.

RT3 used a TV show in her treatment: "I'm a very non-traditional, sort of out of the box therapist, and sometimes what these young males respond to is not a clinically standard

and appropriate type of approach. There's a TV show called "Jackass" where these guys do ridiculous things and often times they're just gross and inappropriate and, in every way, unacceptable behavior. But, I get a better response from "Jackass" than I do almost anything and so I put it on for this young man. . . . The first thing that I noticed, he was watching the screen and not just sitting there, you know, just unaware. He was focusing on the screen and he recognized that the moment was funny and he smiled at the right time. That was my first sign that he was starting to emerge." This practitioner's "out of the box" treatment points to the importance of transgressing disciplinary and normative training to "try things" in order to provide treatments that elicit responses indicative of alertness or arousability or that elicit contextually appropriate responses. This story also shows us that each and every clinical observation a practitioner makes is an additional way for them to collect information that can bring clarity amidst uncertainty of treatment responses to help determine whether patients' responses can be seen as progress in the recovery trajectory.

We report a final example of "thinking in action" and trial and error process as it shows that rehabilitation practitioners are willing to go outside their comfort zones to enable, facilitate, and support patient recovery.

RT11 describes being part of a prayer gathering with a patient's family. She expresses that this activity was out of her comfort zone and had difficulty making sense of the effects this prayer gathering had on the patient. "There was singing, praying, shaking of rattles and drums and things like that. There was two people working with the patient and then two people that worked with his wife. They did breathing work with the wife to release emotional stuff and they did some massage. There were prayers in the Christian and Mayan traditions. Overall, it was a very emotional and amazing experience. The patient had been here for months and had no real response that we could see. So, immediately after that experience, he kind of went into this even deeper sleep, it was like he was knocked out for three days and on the third day when he woke up, he was present. His eyes had changed. He was tracking and showing responsiveness and he just went on this remarkable recovery process that nobody here can explain it. People talk about it and nobody has an explanation. People say it was, he was, a miracle."

In this story of "miracle" recovery, RT11 is not in control of what takes place in the patient's room; she is a participant, not an expert. She can't explain why the patient recovered; her story is cast as a miracle and expresses her own sense making. In the lifeworld of ambiguity and uncertainty that practitioners in DoC navigate, they are explorers, rebels, witnesses of miracles, and improvisers. In the stories presented, practitioners continue trying even though they may not know what might work and why.

"Trying stuff" is an intentional practice. Rehabilitation practitioners "wait to see" how patients respond (experimentation/trial and error), even when they "don't really know why a response is happening." PT7 stated, "we need to try and stimulate [patients'] level of alertness in any way we can." To say that practitioners try to stimulate patients "in any way they" can doesn't mean that 'anything goes.' Rather, practitioners use formal knowledge, prior experiences with previous patients, extrapolation from previous successes, and experimentation (trial and error.) They tell stories to create and enhance meaning making and clinical reasoning. In studies of medicine, this practice is called "doctoring" or "tinkering" [66, 69].

## Discussion

Rehabilitation practitioners are trained in the scientific model of evidence-based medicine (EBM), which includes rational hypothetical-deductive reasoning and logical induction-based algorithms to produce reliable, accurate and valid diagnosis, prognosis, and treatment decisions. In recent years, they are also trained to provide services in person-centered and

culturally competent ways. Medical training and EBM create a cultural framework within which practitioners are cast as experts who know what to do and when to do it. In this framework, empirical knowledge and theory are expected to inform clinical practice with confidence and replicability. Practitioners learn the norms of EBM, the technical tools to assess patients, and the medical language to communicate in. In the EBM literature, uncertainty is viewed as a potential threat to be minimized [71, 72]. EBM has become the canonical framework, and as such it holds epistemic privilege. Some uncertainty is always involved in clinical reasoning. In the clinical lifeworld of rehabilitation practitioners in DoC, ambiguity and uncertainty are omni-present [73, 74]. Practitioners in our study stated that they "don't always know" what to do, that they "try things" in any way they can in order to help patients emerge to consciousness, all the while they second guessed themselves [75], they were unable to explain patient recoveries, experienced assessment discordance (i.e. different practitioners' clinical assessment scores were often not in agreement with each other's) and cognitive dissonance (such as 'double take'). They rarely used language that positioned themselves as knowers [75]. **They tell patient stories in discordance to the cultural frameworks they have been trained in.** They tell patient stories of fluctuation, multiple interpretation, dissonance and doubt, and of transgression from canonical training or treatment. Their stories give us an opportunity to become aware of taken-for-granted practices within the rehabilitation canon that may be otherwise invisible.

In this paper, we discuss the practice of tinkering [69] to make visible the ways practitioners enact clinical and narrative reasoning in this context. Our overall goal is to show the challenges of working in the field of DoC and exhibit practitioners' dedication and creativity to respond to these challenges. In doing so, we show the value that practitioner clinical practices bring to the field of DoC and thus expose the epistemic injustice of treating "thinking in action" as inferior to EBM [76]. We hope that future research and scholarship continue to explicate practitioners' experiences, practices and ways of working with patients in DoC as valuable ways of knowing and doing. Our data and research in clinical practice suggest that "thinking in action" and tinkering are tools/ ways practitioners use in clinical practice. Yet, there is very little in the peer-reviewed literature about the value this practice can offer rehabilitation medicine.

## The practice of tinkering: Clinical reasoning in the midst of ambiguity and uncertainty

Humans reason logically, but also by analogy and through narrative: they use information from familiar areas to link to present situations or problems and tell stories that align with relevant cultural frameworks. This reasoning may be explicit and shareable, or it may be tacit [77]. In our study, practitioners rely on their own clinical expertise, past experiences, and on teammates to interpret patients' responses and make recovery and planning decisions. They make decisions based on judgments, not exactitude [78]; that is, they use a "treasure store of tacit tricks of the trade" such as "a working hypothesis, tradeoffs, risks, intuition" [78]. These "tricks of the trade" are clinical reasoning practices called "doctoring" or "tinkering" [66, 67, 69, 79].

Tinkering is *a way of caring* for patients that involves curiosity, experimentation, struggle, possibly "failing and trying again," being flexible and adapting to complex clinical settings [69]. Tinkering is not an approach where "anything goes." It involves casting oneself into particular narrative roles (rebel, experimenter, observer). The very expression "trying things" that all practitioners used to linguistically express their common practice, suggests that tinkering is part of their everyday lifeworld. Tinkering is how practitioners sometimes have to reason—with creativity and dedication to do what is best for patients, in spite of the ambiguity and uncertainty surrounding them. Tinkering as a practice and as a way of caring, however, is not

generally taught in educational curricula and it is not celebrated as a creative response to caring for complex patients. Tinkering is not perceived as important in the peer reviewed literature since there is a paucity of studies that explicate it as a practice, even though our practitioners clearly use it daily.

**Tinkering and the search for consciousness.** When asked how they made sense of the fluctuation of patients' responses to treatment, practitioners reported that they constantly looked for signs of consciousness. They described this through their stories of looking for "a person being in there," [63] which means observing signs of intention, motivation, or volition that could not be classified as mere bodily reflexes. Considering the high misdiagnosis rates in this population [80–82], efforts to find capacities that signal recovery of volitional abilities, i.e., of consciousness, are significant. 'Looking for a person in the patient in DoC' was the plot, the leitmotif, of many practitioners' patient stories told during interviews. In their stories, practitioners used expressions such as "paying attention to fine nuances" (RT11), "searching for consistency" (SLP1), "trying things" (OT6), and "trying anything to help patients emerge" (PT7). These are examples of tinkering with dedication.

From hermeneutic and narrative perspectives, we recognize interpretation as embedded in clinical reasoning [83–85] and see how it tacitly informs the ways practitioners are trained and work. In their day-to-day work, practitioners actively engage in interpretation. For instance, OTs find patterns in patients' expressions to produce narrative explanations of patients' problems [86]. PTs engage in "piecing clues together to form meaningful wholes" by "a continuing and cyclical process of cue acquisition, hypothesis generation and evaluation of both" [87]. The inclination to find consciousness and, therefore, signs of personhood formed a part of narrative reasoning and "thinking in action" involving piecing together information to make sense of patients' data and circumstances. In other words, while treating patients, practitioners are enacting narrative plots in which they create meaning to make sense of what is happening in their treatments. During interviews, they told stories of their reasoning in which they make sense of their actions. In this paper we have shown how practitioners make sense of (i.e., interpret) the clues patients given them to piece together a meaningful picture of the patient in DoC as a person, rather than as a mere body. In searching for consciousness, practitioners breach the canon of EBM. In looking for consciousness, practitioners tinker with their treatment toolbox. As they tinker, they expose the limitations of the current state of scientific knowledge in the field of DoC. Tinkering, in this sense then, is a clinical reasoning practice that breaches the canon of rehabilitation science. As such, it has the potential to open the field of DoC practice to celebrating practitioners' ways of caring and treating. This may promote exploration and innovation of new treatment modalities and practices.

**Yearning for consistency in the midst of uncertainty.** Practitioners used linguistic expressions such as "is she truly consistent?" and "did I just see that?" signifying their disbelief, ambivalence, lack of certainty and confidence [75] because of patients' fluctuating or inconsistent signs [88]. Philosophers identify yearning for consistency as part of the human condition: in the face of fear and ambiguity, we want certainty [89, 90]. Practitioners may experience self-doubt; they may not know how to make sense of what they are observing. Treating patients from a position of "not knowing" is challenging for practitioners because they "are trained to be experts, [their] job is to know things, to have answers, to educate... Doubt, uncertainty, openness, and reflexivity, however, are essential to avoid stasis, to move rehabilitation in creative directions that best meet the needs of the people and communities we serve" p.141) [68, 91].

In this all too human predicament, practitioners continued to treat, care, and "try things" with patients. They didn't waver, even when not knowing whether or how their interventions impacted their patients. They tried things, observed nuances, adjusted treatments, and didn't

give up. Practitioners marshaled ethics, virtues, experiences, and insights. Clinical reasoning is not just about using the "tricks of the trade" [78]. It is part of the "detective" work of piecing clues together, and of tinkering [66, 67, 69, 79, 87]. Continuous critical review of new evidence and constructive doubting of one's decisions—in other words, practicing with humility—are important elements of being a practitioner in DoC rehabilitation.

## Implications to the field of rehabilitation

In this study, practitioners show us the limitations of a canonical medical culture that focuses on EBM training and valorizes the credentialed professional as the expert. But practicing in the field of DoC is practicing in a borderland [92]. Practitioners are challenged by scientific uncertainty about diagnosis and prognosis and by the ambiguity inherent in treating patients whose responses fluctuate while there is limited evidence to guide treatment decisions. These epistemic limitations have day-to-day consequences for practitioners: they experience lack of confidence and doubt their expertise; they become tinkerers (innovators, improvisers, heroes, rebels, humble observers) in order to respond and treat patients. One implication for the training of practitioners in DoC is to encourage the explicit use of tinkering as a form of clinical reasoning. Uncertainty poses epistemic challenges to EBM. But uncertainty is not necessarily a threat to effectively practice rehabilitation medicine.

Uncertainty may make practitioners uncomfortable and vulnerable and while these are difficult experiences, they open possibilities for creative tinkering that can benefit patients. In the EBM model, uncertainty is a threat. In everyday rehabilitation practice, practitioners' narrative reasoning shows us how uncertainty opens up tinkering. In the field of DoC, where there is epistemic uncertainty, practitioners' ways of knowing and doing are valuable contributions to the treatment process, and "I don't know" is evidence of practicing with humility. Practicing with humility is a strength, not a liability. We hope future studies in the field of medical rehabilitation and DoC in particular will continue to make visible the creative ways that practitioners use to respond to epistemic uncertainty when they care for complex patients. Making visible how practitioners engage in tinkering practices is one way that we contribute to the field of rehabilitation and DoC. Whether tinkering is efficacious to supporting emergence to consciousness for patients in DoC remains unclear and an area of study that needs to be further explored.

While breaching the cannon of EBM may provide an opportunity to promote innovation of new treatment modalities and practices, explicit use of tinkering as a form of clinical reasoning may expose patients to unnecessary risk, or at the very least, expose patients to ineffective interventions. We do not believe the state of the science of tinkering is such that it could be recommended for implementation in a systematic manner. Future research is needed to understand the role of tinkering in tailoring interventions to patients, perhaps by balancing EBM training with ethical clinical practice.

## Limitations

This study involves a small number of participants from two Midwestern rehabilitation facilities with specialized DoC programs and does not represent experiences across facilities and settings. Patients with DoC are often overlooked and not admitted for inpatient rehabilitation, which means our data reflect a vantage point of patients admitted to specialty rehabilitation. Another limitation is recall bias; we were asking practitioners to describe past and current experiences with patients, which may mean that we only heard about the most memorable, frustrating, and surprising experiences. We may have missed opportunities to hear about different types of patients with DoC after TBI. Our study focused on interview narratives and the

stories participants created for the purposes of interviews. We didn't have ethnographic and video data of their clinical encounters. As such, we could not analyze using the tools of conversational analysis and ethnography which would have allowed for more detailed analyses and nuanced discussions of the ways in which practitioners organize their interpretation processes. Future studies should pay more attention to the everyday practices of practitioners by collecting video and ethnographic data of clinical encounters.

## Conclusion

Rehabilitation practitioners who care for patients in DoC work in an environment of ambiguity and uncertainty. Ambiguity exists when there is either no evidence base or there is an imprecise scientific basis to guide diagnoses, prognostication and treatment decisions. Uncertainty occurs when there is high variability in patient responses to treatment and recovery patterns are unpredictable. EBM rehabilitation training curricula do not provide the tools to manage ambiguity, and the diagnostic and prognostic uncertainty of DoC challenges practitioners. The practitioners in our study responded to ambiguity and uncertainty by using their observation skills to monitor nuances in patients' responses that might indicate emerging consciousness. They did so by *searching for consistent* behavioral responses to stimuli as indications that the patient is improving, *observing* fine nuances, and *collaborating* with peers to grapple with unexplained recoveries or stalls in patient progress. While uncertainty raises discomfort, practitioners in our study used "thinking in action" tools such as tinkering to respond to uncertainty in order to care for their patients. They "tried things," used trial and error, worked "outside the box", tweaked things–they tinkered in order to provide optimal care. They sometimes admitted they didn't know why patients recovered the way they did. In admitting they didn't know, they showed their capacity for humility and vulnerability. Practitioners do not simply provide care to patients according to pre-established guidelines; they generate important knowledge by "thinking in action" and tinkering described in this paper. Understanding these practices can lead to new knowledge; practitioners' innovations can generate new insights that can move the science and practice of DoC forward. This study described the innovative ways rehabilitation practitioners deal with ambiguity and uncertainty in working with patients in DoC through tinkering, and as such, opens up the black box of rehabilitation practice.

## Supporting information

**S1 File.**
(PDF)

## Acknowledgments

We would like to thank the rehabilitation practitioners interviewed for this study for their openness to share their patient stories with us. We thank Dr. David A. Stone for his thoughtful editorial feedback and Dr. Alison Cogan for reading an earlier version of this manuscript.

## Author Contributions

**Conceptualization:** Christina Papadimitriou, Trudy Mallinson.

**Data curation:** Jennifer A. Weaver, Ann Guernon, Elyse Walsh.

**Formal analysis:** Christina Papadimitriou, Jennifer A. Weaver, Ann Guernon, Elyse Walsh, Trudy Mallinson.

**Funding acquisition:** Theresa L. Bender Pape.

**Investigation:** Christina Papadimitriou, Ann Guernon, Trudy Mallinson, Theresa L. Bender Pape.

**Methodology:** Christina Papadimitriou, Ann Guernon, Trudy Mallinson, Theresa L. Bender Pape.

**Project administration:** Jennifer A. Weaver, Ann Guernon, Elyse Walsh.

**Resources:** Trudy Mallinson, Theresa L. Bender Pape.

**Supervision:** Christina Papadimitriou, Trudy Mallinson.

**Validation:** Trudy Mallinson.

**Visualization:** Jennifer A. Weaver.

**Writing – original draft:** Christina Papadimitriou, Jennifer A. Weaver, Ann Guernon, Trudy Mallinson.

**Writing – review & editing:** Christina Papadimitriou, Jennifer A. Weaver, Ann Guernon, Trudy Mallinson, Theresa L. Bender Pape.

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
