## [Decision Letter · Decision Letter 0]

26 Aug 2021

PONE-D-21-18770

“Fluctuation is the norm”: Rehabilitation practitioner perspectives on ambiguity and uncertainty in their work with persons in disordered states of consciousness after traumatic brain injury

PLOS ONE

Dear Dr. Papadimitriou,

Thank you for submitting your manuscript to PLOS ONE. After careful consideration, we feel that it has merit but does not fully meet PLOS ONE’s publication criteria as it currently stands. Therefore, we invite you to submit a revised version of the manuscript that addresses the points raised during the review process.

Both reviewers have highlighted some strengths in the manuscript. Yet, there are major concern both at the methodological level and, moreover, at the level of innovation of the findings. The manuscript needs extensive revisions and a careful analysis of all aspects mentioned by the reviewers. Decision about publication will be taken after this revisions. 

We look forward to receiving your revised manuscript.

Kind regards,

Sara Rubinelli

Academic Editor

PLOS ONE

Journal Requirements:

TBP was the PI of these two awards: JWFMRP W81Xwh-16-2-0023; CDMRP W81XWH-14-1-0568

4. Please expand the acronym “CDMRP” (as indicated in your financial disclosure) so that it states the name of your funders in full.

5. Thank you for stating the following in the Acknowledgments/ Funding Section of your manuscript: 

The grant numbers are : JWFMRP W81Xwh-16-2-0023; CDMRP W81XWH-14-1-0568. 

TBP was the PI of these two awards: JWFMRP W81Xwh-16-2-0023; CDMRP W81XWH-14-1-05686. 

6. Thank you for stating the following in your Competing Interests section:  

No competing interests exist

7. In your Data Availability statement, you have not specified where the minimal data set underlying the results described in your manuscript can be found. PLOS defines a study's minimal data set as the underlying data used to reach the conclusions drawn in the manuscript and any additional data required to replicate the reported study findings in their entirety. All PLOS journals require that the minimal data set be made fully available. For more information about our data policy, please see http://journals.plos.org/plosone/s/data-availability.

8. Please include your full ethics statement in the ‘Methods’ section of your manuscript file. In your statement, please include the full name of the IRB or ethics committee who approved or waived your study, as well as whether or not you obtained informed written or verbal consent. If consent was waived for your study, please include this information in your statement as well. 

Reviewers' comments:

Reviewer's Responses to Questions

**Comments to the Author**

1. Is the manuscript technically sound, and do the data support the conclusions?

Reviewer #1: Yes

Reviewer #2: No

2. Has the statistical analysis been performed appropriately and rigorously? 

Reviewer #1: N/A

Reviewer #2: N/A

3. Have the authors made all data underlying the findings in their manuscript fully available?

Reviewer #1: No

Reviewer #2: Yes

4. Is the manuscript presented in an intelligible fashion and written in standard English?

Reviewer #1: Yes

Reviewer #2: Yes

5. Review Comments to the Author

Reviewer #1: Please provide a more detailed definition of severe TBI and DoC. PLOS readers may not have a strong background/understanding of TBI severity/ etiology ect.

The authors cite several articles from the ACRM DoC task force. Would it make sense to discuss this further in your paper? https://acrm.org/acrm-communities/brain-injury/task-forces/disorders-of-consciousness-task-force/

Table 1. Mostly females. Should this be mentioned in the limitations or is this representative of the population?

Please table all exemplar quotes by theme/categories. This makes it easier to read and review.

Please clarify is a qualitative software analysis program was used and if so, which one? NVIVO?

Reviewer #2: The manuscript under review, “Fluctuation is the norm”: Rehabilitation practitioner perspectives on ambiguity and uncertainty in their work with persons in disordered states of consciousness after traumatic brain injury addresses the issues related to uncertainty and unambiguity associated with the rehabilitation science for the improvement of patients in disordered states of consciousness (DoC). The authors have specifically illustrated the importance and limitations of the canonical and scientific procedures adopted by medical practitioners and distinctively provided the importance of alternative tinkering and the canonical breaching methodologies adopted by the practitioners dealing with DoC. Here authors have emphasized that the interpretation and narration of the practitioner’s specific strategy could be his own way of understanding and interpretation. They have addressed this as a major concern for the rehabilitation science dealing with DoC. Overall the manuscript is written in a well narrative manner, but as per my view it lacks the suitability with the journal’s goal, scientific novelty and rigor and is more relevant for the psychological sciences.

Here are the major comments:

1. The major drawback of the manuscript is the unclear conclusion, as the creative tinkering has been an age old practice used for the DoC and is the alternative way in the state of unambiguity and uncertainty by the medical practitioners dealing with DoC. The conclusion drawn here seems very obvious and known to the field.

2. The manuscript need to include some scientific reasoning behind the unambiguity and uncertainty associated with TBI.

3. Again, it would be advantageous to include some scientific reasoning why tinkering methods could be helpful in regaining consciousness.

6. PLOS authors have the option to publish the peer review history of their article (what does this mean?). If published, this will include your full peer review and any attached files.

Reviewer #1: No

Reviewer #2: No

---

## [Decision Letter · Decision Letter 1]

4 Feb 2022

PONE-D-21-18770R1“Fluctuation is the norm”: Rehabilitation practitioner perspectives on ambiguity and uncertainty in their work with persons in disordered states of consciousness after traumatic brain injuryPLOS ONE

Dear Dr. Papadimitriou,

Thank you for submitting your manuscript to PLOS ONE. After careful consideration, we feel that it has merit but does not fully meet PLOS ONE’s publication criteria as it currently stands. Therefore, we invite you to submit a revised version of the manuscript that addresses the points raised during the review process.

Many thanks for the careful revision of the manuscript. One of the reviewers still has some suggestions that I kindly 

ask the author to address.  Please submit your revised manuscript by Mar 21 2022 11:59PM. If you will need more time than this to complete your revisions, please reply to this message or contact the journal office at plosone@plos.org. Please include the following items when submitting your revised manuscript:A rebuttal letter that responds to each point raised by the academic editor and reviewer(s). You should upload this letter as a separate file labeled 'Response to Reviewers'.A marked-up copy of your manuscript that highlights changes made to the original version. You should upload this as a separate file labeled 'Revised Manuscript with Track Changes'.An unmarked version of your revised paper without tracked changes. You should upload this as a separate file labeled 'Manuscript'.If applicable, we recommend that you deposit your laboratory protocols in protocols.io to enhance the reproducibility of your results. Protocols.io assigns your protocol its own identifier (DOI) so that it can be cited independently in the future. For instructions see: https://journals.plos.org/plosone/s/submission-guidelines#loc-laboratory-protocols. Additionally, PLOS ONE offers an option for publishing peer-reviewed Lab Protocol articles, which describe protocols hosted on protocols.io. Read more information on sharing protocols at https://plos.org/protocols?utm_medium=editorial-email&utm_source=authorletters&utm_campaign=protocols.

We look forward to receiving your revised manuscript.

Kind regards,

Sara Rubinelli

Academic Editor

PLOS ONE

Journal Requirements:

Reviewers' comments:

Reviewer's Responses to Questions

**Comments to the Author**

1. If the authors have adequately addressed your comments raised in a previous round of review and you feel that this manuscript is now acceptable for publication, you may indicate that here to bypass the “Comments to the Author” section, enter your conflict of interest statement in the “Confidential to Editor” section, and submit your "Accept" recommendation.

Reviewer #1: (No Response)

Reviewer #2: All comments have been addressed

2. Is the manuscript technically sound, and do the data support the conclusions?

Reviewer #1: Yes

Reviewer #2: Yes

3. Has the statistical analysis been performed appropriately and rigorously? 

Reviewer #1: Yes

Reviewer #2: N/A

4. Have the authors made all data underlying the findings in their manuscript fully available?

Reviewer #1: Yes

Reviewer #2: Yes

5. Is the manuscript presented in an intelligible fashion and written in standard English?

Reviewer #1: Yes

Reviewer #2: Yes

6. Review Comments to the Author

Reviewer #1: Well done! Thank you for your revisions and for your response to the reviewer's comments. This is a significant contribution to the existing literature.

Reviewer #2: The authors of the manuscript (“Fluctuation is the norm”: Rehabilitation practitioner perspectives on ambiguity and uncertainty in their work with persons in disordered states of consciousness after traumatic brain injury”) have provided satisfactory answers to the questions. The authors are still advised to provide additional information for the following comments before the final submission.

1. Can there be any potential disadvantages of the tinkering? Please throw some light on the potential ways these practices can be implemented in a systematic manner and what are the potential limitations.

7. PLOS authors have the option to publish the peer review history of their article (what does this mean?). If published, this will include your full peer review and any attached files.

Reviewer #1: No

Reviewer #2: No

---

## [Author Response · Author response to Decision Letter 1]

2 Mar 2022

We have uploaded a letter responding to reviewers' comments.

---

## [Decision Letter · Decision Letter 2]

5 Apr 2022

“Fluctuation is the norm”: Rehabilitation practitioner perspectives on ambiguity and uncertainty in their work with persons in disordered states of consciousness after traumatic brain injury

PONE-D-21-18770R2

Dear Dr. Papadimitriou,

We’re pleased to inform you that your manuscript has been judged scientifically suitable for publication and will be formally accepted for publication once it meets all outstanding technical requirements.

Kind regards,

Sara Rubinelli

Academic Editor

PLOS ONE

Additional Editor Comments (optional):

Reviewers' comments:

Reviewer's Responses to Questions

**Comments to the Author**

1. If the authors have adequately addressed your comments raised in a previous round of review and you feel that this manuscript is now acceptable for publication, you may indicate that here to bypass the “Comments to the Author” section, enter your conflict of interest statement in the “Confidential to Editor” section, and submit your "Accept" recommendation.

Reviewer #1: All comments have been addressed

Reviewer #2: All comments have been addressed

2. Is the manuscript technically sound, and do the data support the conclusions?

Reviewer #1: Yes

Reviewer #2: Yes

3. Has the statistical analysis been performed appropriately and rigorously? 

Reviewer #1: Yes

Reviewer #2: N/A

4. Have the authors made all data underlying the findings in their manuscript fully available?

Reviewer #1: Yes

Reviewer #2: Yes

5. Is the manuscript presented in an intelligible fashion and written in standard English?

Reviewer #1: Yes

Reviewer #2: Yes

6. Review Comments to the Author

Reviewer #1: Recommend Accept. This is important and timely research. Will be of interest to the PLOS community.

Reviewer #2: The manuscript in the present form looks great and as per my opinion is suitable for the publication.

7. PLOS authors have the option to publish the peer review history of their article (what does this mean?). If published, this will include your full peer review and any attached files.

Reviewer #1: No

Reviewer #2: **Yes: **Hemendra J. Vekaria

---

## [Editor Report · Acceptance letter]

12 Apr 2022

PONE-D-21-18770R2 

“Fluctuation is the norm”: Rehabilitation practitioner perspectives on ambiguity and uncertainty in their work with persons in disordered states of consciousness after traumatic brain injury 

Dear Dr. Papadimitriou:

I'm pleased to inform you that your manuscript has been deemed suitable for publication in PLOS ONE. Congratulations! Your manuscript is now with our production department. 

Kind regards, 

on behalf of

Dr. Sara Rubinelli 

Academic Editor

PLOS ONE